# What Do We Know about Theory of Mind Impairment in Parkinson’s Disease?

**DOI:** 10.3390/bs11100130

**Published:** 2021-09-24

**Authors:** Clara Trompeta, Beatriz Fernández Rodríguez, Carmen Gasca-Salas

**Affiliations:** 1HM CINAC MADRID (Centro Integral de Neurociencias Abarca Campal), Hospital Universitario HM Puerta del Sur, HM Hospitales, 28938 Móstoles, Spain; claratrompetau@gmail.com (C.T.); beabeafdez@gmail.com (B.F.R.); 2Network Center for Biomedical Research on Neurodegenerative Diseases (CIBERNED), Instituto Carlos III, 28031 Madrid, Spain

**Keywords:** Parkinson’s disease, theory of mind (ToM), cognitive ToM, affective ToM, neuroimage, mild cognitive impairment, cognitive domains, depression

## Abstract

Theory of mind (ToM) is a social cognitive skill that involves the ability to attribute mental states to self and others (what they think (cognitive ToM) and feel (affective ToM)). We aim to provide an overview of previous knowledge of ToM in Parkinson’s disease (PD). In the last few years more attention has been paid to the study of this construct as a non-motor manifestation of PD. In advanced stages, both components of ToM (cognitive and affective) are commonly impaired, although in early PD results remain controversial. Executive dysfunction correlates with ToM deficits and other cognitive domains such as language and visuospatial function have also been related to ToM. Recent studies have demonstrated that PD patients with mild cognitive impairment show ToM deficits more frequently in comparison with cognitively normal PD patients. In addition to the heterogeneity of ToM tests administered in different studies, depression and dopaminergic medication may also be acting as confounding factors, but there are still insufficient data to support this. Neuroimaging studies conducted to understand the underlying networks of cognitive and affective ToM deficits in PD are lacking. The study of ToM deficit in PD continues to be important, as this may worsen quality of life and favor social stigma. Future studies should be considered, including assessment of the patients’ cognitive state, associated mood disorders, and the role of dopaminergic deficit.

## 1. Introduction

Non-motor symptoms are common in Parkinson’s disease (PD) [1]. In this regard, cognitive impairment and dementia are frequent entities, especially in patients with long disease duration [2,3]. A wide variety of cognitive deficits have been reported in PD, even early in the disease and with different development over time. Classically, the cognitive deficit profile in PD has been linked to a pattern with more executive deficits caused by dopaminergic fronto-striatal network dysfunction in early stages [4]. Nevertheless, patients exhibiting posterior cortical deficits show a higher risk of conversion to dementia [5]. There is also a syndrome representing a stage between normal aging and dementia in PD without loss of functionality (mild cognitive impairment in PD- MCI-PD) [5,6,7], and this is considered a risk factor for dementia [8]. When PD patients develop dementia, they show a greater degree and diversity of cognitive deficits (affecting aspects such as memory, visuospatial function, attention, and executive function) severe enough to affect the daily life routine [4]. PD patients can also suffer from subjective cognitive decline (SCD)—a stage that represents an increased risk for cognitive decline in some studies—where cognitive deficits may be noticed by the patient, family or clinician, but might be too slight to be noticed in a formal cognitive evaluation [9]. Social cognition (SC) is a cognitive domain that can also be affected in PD patients but has been less commonly studied [10]. This domain implies a wide variety of cognitive functions directed to perception, processing, and interpretation of social information, which allows us to manage social relationships with those around us appropriately [11,12]. Such is the importance of the study of SC as a cognitive domain that the Diagnostic and Statistical Manual for Mental Disorders [13] considers it to be one of the six key domains of mental function that should be evaluated in MCI and dementia. Although there is some disagreement with respect to the components of SC, as some of them share similar characteristics [14], the most important and most widely studied in PD are theory of mind (ToM), empathy, and social perception [15].

## 2. Theory of Mind

ToM may be defined as the ability to understand and predict other people’s behavior, knowledge, intentions, emotions, and beliefs [16]. This term was first proposed by Premack and Woodruff in 1978 based on primate studies [17] and shortly afterwards reported by Baron-Cohen in studies on children with autism [18]. Later, ToM was investigated in healthy population, people with schizophrenia [19,20], and with regard to neurodegenerative disorders that share the label of ‘frontotemporal dysfunction syndromes’ such as frontotemporal dementia (FTD) [21] and more recently in amyotrophic lateral sclerosis (ALS) [22]. In this sense, ALS patients show problems in both affective and cognitive ToM (two parts of ToM explained in more depth below) even in early stages of the disease [23]. Affective ToM deficits in ALS have been related to the disruption of the frontotemporoparietal network [24,25], while cognitive ToM deficits have been linked to a separate network. In this regard, one study found bilateral hypometabolism in the dorsomedial and dorsolateral prefrontal cortex and in the supplementary motor area associated with poor cognitive ToM performance in this disease [26]. Likewise, moderate impairments in overall ToM have also been found in multiple sclerosis (MS) an immune-mediated demyelinating disease of the central nervous system (with no differences between its subtypes). This impairment might be explained by white matter damage in MS [27]. In general, all these data support the idea that this construct may also be an acquired disorder and is not only developmental.

ToM consists of a complex metacognitive skill set that begins to develop during brain maturation in infancy and continues during the growth stage [16]. Although the concept was initially described as a single entity [28], more recent neuropsychological and neuroimaging studies have facilitated subdivision into different types, namely cognitive and affective ToM [29]. The cognitive aspect is the capacity to understand what others think, intend, and believe [30], whereas the affective part is the process by which someone can understand what others feel [31]. Some authors consider affective ToM to be akin to cognitive empathy, while affective empathy refers to the emotional response to the perceived situation of another [12,15,32].

Another way to categorize ToM is the first-grade belief or the ability that allows the subject to distinguish his/her own beliefs or emotions from those of others; second-grade belief which allows him/her to know that someone thinks that a third person believes or feels something, and *faux pas* or the ability to identify social gaffes, deception and sarcasm, which are also considered to be part of other advanced mentalist abilities. First order ToM develops between three and four years of age. Second order ToM, *faux pas*, deception and sarcasm are developed from five to ten years, approximately [20].

## 3. General Neural Bases of Theory of Mind

Due to the complexity of this social process, ToM cannot be attributed from a neuroanatomical point of view to a single structure, but to a network of brain functions [33]. The medial prefrontal cortex (MPFC) contributes both to self-referential thinking and to imagining the perspective of others and the temporoparietal junction (TPJ) is essential to understand that another person may think or believe something different from yourself. Both areas have been described as key regions in ToM [34]. Previous neuroimaging ToM studies in the general population including positron emission tomography (PET) and functional magnetic resonance imaging (fMRI) showed high variability, likely due to different methodological approaches. According to a review by Carrington et al., the most specific brain regions involved in ToM tasks include the MPFC and orbitofrontal cortex (OFC), with activity reported in 93% of studies, as well as the lateral prefrontal cortex, reported in 35%, the anterior cingulate cortex (ACC) and para-cingulate (PCC), in 55% and TPJ in 58%. As a conclusion, they identified “core” ToM regions from the previous neuroimaging studies reviewed. These were: MPFC/OFC, superior temporal sulcus (STS), TPJ, ACC, and PCC, whereas the amygdala was less frequently activated [35]. More recently, it has been discussed [36] how TPJ and posterior STS have been rearranged in humans with the evolution of social cognition, and how this ability seems to be asymmetrically located in the right hemisphere. Some of the areas involved in ToM are part of the default mode network (DMN), a resting-state network including the posterior cingulate cortex and adjacent precuneus, mesial and inferior temporal lobes, inferior parietal lobe and MPFC. DMN is responsible for the self-referential introspective state [37] and autobiographical memory [38], but also for thoughts about self versus others and ToM [39].

It has also been speculated that the cognitive and affective components of the ToM involve different networks. Based on previous work [40,41,42] Abu-Akel [43] proposed a cognitive ToM network, involving the dorsolateral (DLPFC) and dorsomedial prefrontal cortex (DMPFC), dorsal ACC, dorsal temporal pole and dorsal striatum; and an affective ToM circuit including the inferior lateral frontal cortex, ventromedial prefrontal cortex (VMPFC), OFC, amygdala, and the ventral part of the striatum, temporal lobe and ACC.

## 4. Anatomical and Functional Correlates of Theory of Mind in Parkinson’s Disease

The dopamine depletion characterizing PD impacts the basal ganglia networks involving not only the motor circuit [44] but also the associative and the limbic circuits [45] (see Figure 1).

In this regard, it has been hypothesized that the progressive loss of striatal denervation in PD could initially lead to dysfunction of the associative circuit and therefore of cognitive ToM, associated with other symptoms such as executive dysfunction. In more advanced stages, the limbic circuit dysfunction would affect affective ToM and cause mood disorders [42,46,47,48].

Few neuroimaging studies [49,50,51] (see Table 1) using different methodological approaches have searched for anatomical correlates of ToM performance in PD. Grey matter voxel-based morphometry analysis demonstrated a correlation between cognitive ToM performance, and left precentral and postcentral gyrus, anterior cingulate gyrus, middle frontal gyrus and inferior frontal gyrus volumes [50]. The same study shows white matter volume decreased in the superior longitudinal fasciculus (adjacent to the parietal lobe), and white matter adjacent to the frontal lobe correlated with cognitive ToM performance. After controlling for executive functions (EF), the relationship between ToM deficit remained significant for white matter areas adjacent to the precuneus and the parietal lobe, but no correlations existed between ToM and grey matter volume [50], suggesting that EF could be part of the ToM or act as a confounding factor.

In a recent study, affective ToM performance positively correlated with cortical metabolism in the insula and superior temporal gyrus. According to I-123 ioflupane single photon emission computed tomography (123 I-FP-CIT-SPECT), ToM scores did not correlate with basal ganglia and cortical dopaminergic function. This technique was also used as a proxy marker of serotoninergic impairment revealing a negative correlation with thalamic specific binding ratio. Cholinergic function was indirectly measured by quantitative electroencephalography (posterior and anterior Theta/Alpha power ratio as a proxy of cholinergic tone) [51]. These findings suggest a lack of dopamine involvement in ToM. However, 123 I-FP-CIT-SPECT has limited spatial resolution and the authors only assessed affective ToM [51].

Finally, a study in PD patients undergoing subthalamic nucleus deep brain stimulation (STN-DBS) showed worsening of affective ToM that correlated with brain hypometabolism (according to Fluorodeoxyglucose-PET) in the bilateral cingulate gyrus, bilateral middle frontal gyrus, left middle frontal gyrus, temporal lobe (fusiform gyrus), bilateral parietal lobe and bilateral occipital lobe. In addition, the ToM impairment correlated with hypermetabolism in the left superior temporal gyrus and bilateral inferior frontal gyrus [49].

## 5. Clinical Assessment of ToM in PD

A wide variety of tools have been used to evaluate ToM in PD [12,14,52]. Some of the tests have shown good reliability and sensitivity in other neurodegenerative disorders such as the behavioral variant of frontotemporal dementia (bvFTD) [15]. Others were used first in autism or schizophrenia [20,53]. Depending on task demands, cognitive, affective or both ToM subtypes are assessed [52]. Some of the most commonly used tools to assess this cognitive construct based on cognitive and affective subtypes are:Cognitive ToM:

*False-belief task* [54]: in which participants have to understand that a character in a story has a belief which is different to reality and to their own (first order belief). Newer versions in which affective ToM is also evaluated have been developed. For example, the Italian version of the *Yoni task* [55], a sensitive tool for detecting different dimensions of ToM impairment.

*Strange Stories Test* [53]: a task in which the patient can understand the character’s behavior by attributing a mental state to them.

*Advanced Test of ToM* (AT) [53]: a task to investigate the ability to attribute mental states to others.

*ToM stories test* [56]: in which first order belief, second order belief, cooperation between characters and one character deceiving another are evaluated.

Affective ToM:

*Reading the Mind in The Eyes Test (RMET)* [57]: it is one of the most widely used instruments in affective ToM to understand what a person in a photograph is feeling. Two scores can be obtained: the emotion score (the total number of items identified correctly by the patient) and the gender score (by attributing correctly the gender of the photography actor). A percentage of the correct responses can be calculated later in both cases.

*The Emotion Attribution Task (EAT)* [58]: this is a more recent task that consists of short stories describing emotional situations.

Cognitive and affective ToM:

*Faux-Pas test* [57]: this tests whether the subject can understand gaffes or an action that is a social mistake or impolite.

*The Awareness of Social Inference Test (TASIT)* [59]: these assess the ability to perceive social cues that are performed in realistic scenarios.

## 6. Theory of Mind Performance in Parkinson’s Disease Patients

### 6.1. Main Findings and Potential Confounders

ToM has been studied in PD for the last 20 years. The first study found worse performance of a ToM task (false-belief task) in 11 non-demented patients in comparison with 8 elderly controls and 9 university-aged control participants [28]. However, in addition to the low sample size, some information such as demographic data (e.g., disease duration or dopaminergic medication doses) was lacking. Later, Péron et al. [52] compared 17 early PD patients with 27 advanced PD and 26 age-matched healthy controls (HC). There were no differences between early PD patients (in the medicated and non-medicated conditions) versus HC group on either cognitive (Faux Pas test) or affective (RMET test) ToM performance. However, more advanced patients showed deficits in cognitive ToM but not in affective ToM.

The first systematic review about ToM in basal ganglia disorders included ten studies on PD, showing that cognitive and affective ToM were impaired in advanced stages of the disease, whereas results in early stages remained controversial [31].

Roca et al. [60] were the first to report deficits in cognitive ToM in early PD patients with and without dopaminergic medication. They compared a homogeneous sample in terms of disease duration and disease severity of 36 early PD patients (16 under medication effect and 20 unmedicated patients) with 35 matched HC subjects and found cognitive ToM dysfunction (according to Faux Pas test) in early stages of the disease but not in affective ToM (measured by RMET) in comparison with HC and regardless of medication state. However, another study found that both affective (modified Italian version of EAT test) and cognitive (AT test) ToM tasks were already impaired in 33 early medicated and non-demented PD patients in comparison with 33 age- and education-matched HC [61], although disease duration was longer than in the previous study (mean: 6.8 ± 4.7 years vs. 1.7 ± 1.6 years). A subsequent meta-analysis including 18 studies with 487 non-demented medicated PD patients and 459 HC, found a significant impairment in ToM in both cognitive and affective ToM, the cognitive component being more affected than the affective one. These deficits were less severe in early PD compared with more advanced patients. Nevertheless, only five of the studies included compared both ToM components. In addition, the effect of dopaminergic treatment in ToM performance could not be evaluated due to insufficient data [62]. A more recent meta-analysis including a larger number of studies (38 studies with 1014 non-demented medicated PD patients) showed that PD patients had significant and moderate-sized difficulties in both cognitive and affective ToM (or cognitive empathy) compared with an age-matched HC group (*n* = 921) [12].

Focused on early-onset medicated PD patients with disease duration less than 5 years (*n* = 25), a recent study found poor performance in several SC tests, among which affective (with RMET) but not cognitive ToM was also evaluated. Half of this sample showed a moderate to severe motor state [63].

Finally, there is a single longitudinal study in 16 drug-naïve newly diagnosed PD patients finding baseline impairment in cognitive ToM (according to Faux Pas test) but not in affective ToM (assessed with EAT) [64]. These findings favor the idea of an early deterioration of cognitive aspects of ToM due to an early degeneration of dorsolateral-prefrontal-striatal circuitry in PD. Furthermore, they found a significant improvement in the Faux Pas test results in the PD group three months and one-year after dopaminergic medication was started.

Moreover, some studies found that mood disorders such as depression influenced ToM performance in PD [65], whereas another study did not support that finding [66].

### 6.2. Relationship with Other Cognitive Functions and General Cognitive Function

It has been suggested that SC impairment (and hence ToM impairment) could be either a primary disorder or secondary to another related deficit [15]. Since the human being is so social, the aspects related to cognition and behavior are in some way social, erasing the boundaries between ‘social’ and ‘non-social’ cognitive aspects [33]. Therefore, processes such as perception, memory, attention, reasoning, or decision making are functions underlying social abilities and the first cannot be understood without the second.

Some authors assume that frontostriatal deficits in PD could affect not only EF but also ToM as they could share the same circuitry. In this regard, most studies found an association between EF and cognitive ToM performance [28,61,65]. Peron et al. [52] found these relations in advanced PD but not in newly diagnosed patients, suggesting that ToM impairment in advanced disease is due to a general cognitive decline but cannot be related to the dopaminergic denervation. A subsequent meta-analysis found a significant relationship between ToM impairment and EF deficits (measured with verbal fluency) [62]. This was recently supported by a study in which through a reduction in the level of inhibition (a measure of EF) the cognitive ToM task became unimpaired in non-demented PD patients [67].

On the other hand, a few studies show an association between ToM and other cognitive domains. One study investigated the relationship between language and ToM in PD [68] finding an association between second-order mental state attribution and pragmatic abilities (the use of language depending on context) in a group of 11 early to moderate PD patients in whom those abilities were impaired. A relationship between affective ToM and visuospatial function was also shown, arguing that the second could explain the first even in early stages of PD [69]. Later, Alonso-Recio et al., [14] found an association between affective ToM (measured with RMET) and processing speed, memory, and EF.

With respect to the cognitive state, most studies have included “non-demented” PD patients. This term includes MCI and cognitively normal (CN) patients as a single group, but their cognitive performance is different. However, two recent studies included them as separate groups. The first one observed that PD-MCI patients (diagnosed according to the Montreal Cognitive Assessment (MoCA) score < 26) performed worse than PD-CN and a HC group in affective ToM (measured with RMET). However, cognitive ToM was not evaluated [14]. The second study included 109 medicated and newly PD patients with and without MCI (PD-MCI cognitive diagnosis performed according to MoCA score < 26) and found that PD patients performed significantly worse than HC in cognitive and affective ToM. Isolated social cognition impairment (SCI) was 3.5 times more frequent than PD-MCI diagnosis in their sample and a sub-group of PD patients without MCI showed isolated SCI. They also showed an influence of EF (assessed with the frontal assessment battery, FAB) on the ToM task [66]. Nonetheless, both studies used the level I (abbreviated assessment) diagnostic category of the Movement Disorder Society Task Force in the classification of cognitive impairment in PD, based on a scale of global cognitive abilities such as the MoCA. This is less sensitive than the level II diagnostic category that allows the differentiation between PD-MCI profiles by including a comprehensive neuropsychological assessment [70].

## 7. Discussion and Future Challenges

SC and especially ToM are essential processes that allow the correct management of interpersonal relationships [12]. More specifically, cognitive ToM includes the capacity to understand what others think, believe and are trying to do and affective ToM includes the ability to understand what others feel [18,31]. First grade belief, second grade belief, faux pas, deception and sarcasms are also processes that make up the ToM concept and are developed during childhood [20]. They have been studied in different populations, such as patients with autism or schizophrenia, and more recently in neurodegenerative disorders including PD [18,19,20,21,22,23,24,25,26,27].

Such is the importance of the evaluation of SC in general, and of ToM in particular, in neurodegenerative disorders, that they are considered a domain that should be taken into account for the diagnosis of MCI and dementia [13]. Non-motor manifestations affect the majority of PD patients, in whom cognitive dysfunction is very common [2,9]. Initial studies in PD considered that cognitive ToM and affective ToM are both affected in advanced PD patients [31] but in early stages the results have not been consistent, likely due to methodological differences such as different ToM batteries and confounding factors like disease duration and onset, the patient’s medication status, as well as the influence of other cognitive functions/cognitive state and mood disorders.

It has been suggested that the anatomical regions involved in ToM are integrated into two functionally dissociated circuits according to a neurobiological model. The cognitive ToM network mainly includes the dorsal ACC, the DMPFC, and the dorsal striatum/caudate. On the other hand, the affective ToM network primarily comprises the VMPFC and OFC, the ventral ACC, the amygdala, and the ventral striatum [43]. These are consistent with the basal ganglia associative and limbic networks respectively, which are typically impaired in PD patients as the disease progresses [47,48]. However, the number of neuroimaging studies assessing ToM in PD patients is low and they do not assess both affective and cognitive components. On the other hand, these studies support an association between ToM impairment and white and grey matter volume decrease as well as hypometabolism in several brain areas [49,50,51].

Early nigrostriatal degeneration in PD would lead to dysfunction of the associative circuit and therefore EF and cognitive ToM. In more advanced stages mesolimbic impairment would explain the mood and affective ToM problems [31]. The impairment of EF early in the disease has been well established, and correlations between cognitive ToM and EF have been demonstrated in several studies [28,61,62,65,67]. However, according to this hypothesis, mood disorders are not likely to be present in early stages of the disease, even though it is well known that mood problems such as depression can be present even in the prodromal stage [71]. Extensive, more recent studies support impairment of both cognitive and affective ToM in both early and advanced PD [12,62,66]. Regarding age at onset, only one study assessed ToM in early-onset PD, finding impairment of affective ToM, but cognitive ToM has not been studied in this context [63].

The influence of depression disorders on ToM performance is controversial in the general population [72], and the same is the case in PD. One study found that PD patients with higher depression scores performed worse in the ToM task than those with lower depression scores [65]. Another study found no influence of this mood disorder on patient performance [66]. Nonetheless, those subjects showed low levels of depression. Therefore, it was not possible to confirm that moderate to severe depression does not influence ToM performance. Moreover, the influence of other mood disorders such as apathy, anxiety, or impulse control in ToM performance in PD has not been studied to date. Adding neuropsychiatric questionnaires for the evaluation of mood disorders will help us to understand if they play any role in ToM in PD. In addition, as no standardized battery has been established to evaluate SC in PD, the different ToM tests administered may contribute to some heterogeneous findings among the studies.

The cognitive state also influences ToM outcomes. Only two recent studies differentiated among CN and MCI PD patients [14,66] finding that ToM impairment is 3.5 times more common in PD-MCI, and also identifying a subgroup of patients with SCI but without an MCI diagnosis [66]. This supports the need to evaluate PD patients using a more comprehensive neuropsychological assessment that allows a better differentiation between cognitive profiles, including ToM as a separate domain.

According to the basal ganglia circuits, it has been suggested that dopamine depletion in PD could explain ToM impairment. However, most studies did not find differences between medicated and non-medicated (with dopaminergic drugs) patients [52,60]. On the other hand, a longitudinal study shows a better performance in de novo PD patients after starting dopaminergic medication [64] although a practice effect cannot be excluded. Nevertheless, there are not enough data about the interplay between dopaminergic medication and disease severity in ToM performance. A single study did not find any correlation between dopaminergic function and ToM scores according to 123 I-FP-CIT-SPECT. However, this has limited spatial resolution and the authors assessed affective but not cognitive ToM [51]. That study also suggested a potential serotoninergic involvement, but this should be confirmed with serotonin transporter imaging. Thus, the role of other neurotransmitters such as serotonin and oxytocin, associated with social functioning, may also contribute to ToM, as previously detected in autism and schizophrenia studies [43,73,74]. Finally, there is a potential relationship between noradrenaline and ToM in the general population [75] and this neurotransmitter is reduced in PD [9]. Noradrenaline relates to cognitive dysfunction (especially with executive dysfunction and attention) [9] as well as depression and apathy in PD [76]. However, no studies have shown a link between norepinephrine loss and locus coeruleus degeneration with ToM deficits in this neurodegenerative disease.

There are some limitations in our review. Although we added the most relevant publications of ToM in PD, this is not a systematic review, and this prevents us from performing an analysis of paper quantity/quality. Nonetheless, we aimed to report a narrative review including other aspects such as the comprehensive background to the ToM concept, suggested circuitries affected in PD or test descriptions. On the other hand, we did not include the study of facial emotion recognition (FER, previously included in few ToM studies), but this is not considered a part of the ToM construct as reported previously in the scientific literature [15,77,78,79].

In conclusion, ToM, especially the cognitive subcomponent, is impaired in PD. Whereas in advanced PD, both cognitive and affective ToM are affected, previous findings on early patients are not uniform, likely due to several confounding factors. The study of ToM in PD is important as these deficits worsen patients’ quality of life and could increase the patients´ social isolation. Future investigations should include full neuropsychological batteries allowing for the study of all cognitive domains in CN and MCI patients, as well as the evaluation of depressive patients, and analysis of the interaction between dopaminergic medication and disease severity. More neuroimaging studies that may explain the underlying networks of ToM deficits in PD as well as dopamine and serotonin transporter imaging studies are warranted.

## Figures and Tables

**Figure 1 behavsci-11-00130-f001:**
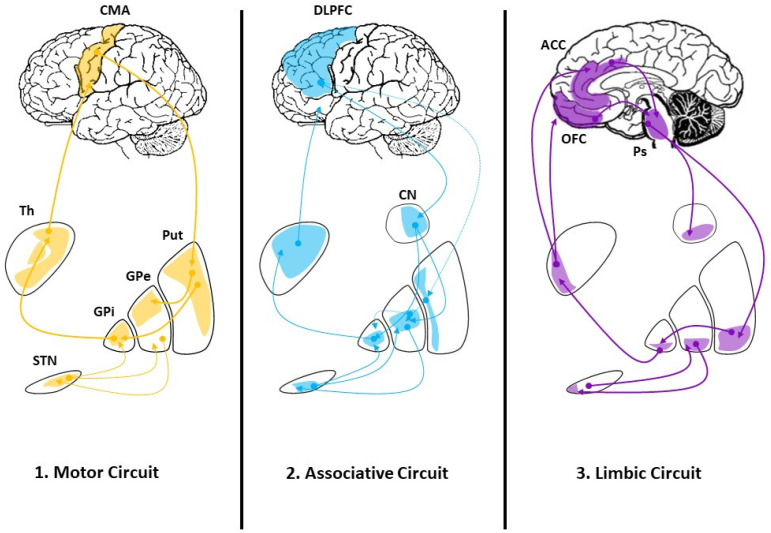
Schematic diagram of the main functional division of the cortico-basal ganglia connections sub-divided according to the main cortical projection areas. (**1**). The motor circuit: cortical motor areas. (**2**). The associative circuit: dorsolateral prefrontal cortex. (**3**). The limbic circuit: orbitofrontal cortex and anterior cingulate cortex (modified from Obeso et al. [45]). CMA: cortical motor areas; Th: thalamus; Put: putamen; GPe: globus pallidus pars externa; GPi: globus pallidus pars interna; STN: subthalamic nucleus; DLPFC: dorsolateral prefrontal cortex; CN: caudate nucleus; ACC: anterior cingulate cortex; OFC: orbitofrontal cortex, Ps: pons.

**Table 1 behavsci-11-00130-t001:** Neuroimaging studies in Parkinson’s disease and ToM.

	Sample; Mean Age (Years)	Motor Scale	Disease Duration	LEDD ^6^	ToM Test	Anatomical Correlate
Péron et al. 2010 18 FDG PET ^1^	PD ^2^: 13; 53.3 (8.5) HC ^3^: 13	UPDRS ^4^ OFF pre STN-DBS: 31.4 (12.2) ON pre STN-DBS: 8.8 (4.5) OFF MED post STN-DBS: 14.1 (7.4) ON MED post STN-DBS: 6.1 (4.0)	10.5 (3.6)	Pre STN-DBS ^5^: 1081.1 (605.3) Post STN-DBS:625.8 (600.9)	RMET	Pre VS. Post STN-DBS ON STIM * Hypometabolism: bilateral anterior cingulate gyrus & L ^7^ superior frontal gyrus Hypermetabolism: bilateral cerebellum & R ^8^ inferior parietal lobule Correlation between decreased glucose metabolism and impaired TOM ** Bilateral cingulate gyrus, R middle frontal gyrus, L middle frontal gyrus, temporal lobe, bilateral parietal lobe, bilateral occipital lobe Correlation between increased glucose metabolism and impaired TOM ** L superior temporal gyrus, L inferior frontal gyrus, R inferior frontal gyrus
Díez-Cirarda et al. 2015 MR: GM ^9^ analysis VBM ^10^ WM ^11^ analysis: TBSS ^12^	PD: 37; 67.97 (6.17) HC: 15; 65.07 (7.01)	UPDRS: 21.72 (10.29)	6.96 (5.61)	808.59 (536.81)	Strange Stories Task	Correlations between ToM and GM volume in PD *** L precentral and postcentral gyrus, anterior cingulate gyrus, middle frontal gyrus, inferior frontal gyrus Correlations between ToM and GM volume in PD, controlling for executive functions: no clusters Correlations between ToM and WM in PD: **** FA ^13^: R superior longitudinal fasciculus MD ^14^: L superior longitudinal fasciculus, left external capsule RD ^15^: L superior longitudinal fasciculus Correlations between ToM and WM in PD, controlling for executive functions: FA: R superior longitudinal fasciculus *** MD: L superior longitudinal fasciculus, L inferior longitudinal fasciculus **** RD: R corticospinal tract, L superior longitudinal fasciculus, L inferior longitudinal fasciculus ***
Orso et al. 2020 18 FDG PET 123-FP-CIT SPECT ^16^	PD: 30; 73.39 (8.93) HC: 60; 70.1 (10.9)	MDS-UPDRS III ^17^: 20.65 (7.6)	De novo	Drug-naïve	RMET	Positive correlation with cortical metabolism in the insula and superior temporal gyrus. Serotoninergic function: negative correlation with thalamus SBR ^18^ in the LAH ^19^. No correlation with: basal ganglia dopaminergic function, dopaminergic function in the cortical RMET ^20^-related VOI ^21^

^1^ 18 FDG PET: (18)F-fluorodeoxyglucose positron emission tomography; ^2^ PD: Parkinson’s disease; ^3^ HC: healthy controls; ^4^ UPDRS: Unified Parkinson’s disease rating scale; ^5^ STN-DBS: subthalamic nucleus deep brain stimulation; ^6^ LEDD: levodopa equivalent daily dose; ^7^ L: left; ^8^ R: right;^9^ GM: grey matter; ^10^ VBM: voxel based morphometry; ^11^ WM: white matter; ^12^ TBSS: Tract-based spatial statistics analysis; ^13^ FA: fractional anisotropy, ^14^ MD: mean diffusivity; ^15^ RD: radial diffusivity; ^16^ 123-FP-CIT SPECT: I-123 ioflupane single photon emission computed tomography; ^17^ MDS-UPDRS III: Movement Disorders Society unified Parkinson’s disease rating scale III; ^18^ SBR: specific binding ratio; ^19^ LAH: less affected hemisphere; ^20^ RMET: Reading the Mind in the Eyes Task; ^21^ VOI: volume of interest. Age is reported as mean (standard deviation). * *p* < 0.001, with multiple comparison correction. ** *p* < 0.005, with multiple comparison correction. *** *p* < 0.001 uncorrected. **** *p* < 0.05 FWE-corrected.

## Data Availability

Not applicable.

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
