# Peer review of "What Do We Know about Theory of Mind Impairment in Parkinson’s Disease?"

_behavsci, 2021, doi:10.3390/bs11100130_

Round 1

Reviewer 1 Report

Good summary of an important neuropsychological concept in the field of PD. Highlights social cognitive deficits, that are an often overlooked but are an important aspect of cognitive impairment in PD.

Article is well written, comprehensive but it could be made more meaningful if the authors expand it to include concepts of empathy and facial recognition, that also form an important aspect of social cognitive deficits noted in PD.

Also, it would be meaningful to compare ToM deficits noted in PD to other diseases like those in ALS, or MS, as the underlying circuitry affected is different in all these conditions.

Author Response

Reviewer 1. 

Comments to the Author:

  1. Good summary of an important neuropsychological concept in the field of PD. Highlights social cognitive deficits, that are an often overlooked but are an important aspect of cognitive impairment in PD.

We thank the reviewer for the positive comments.

  1. Article is well written, comprehensive but it could be made more meaningful if the authors expand it to include concepts of empathy and facial recognition, that also form an important aspect of social cognitive deficits noted in PD.

As social cognition has diverse components, we have focused our review specifically on ToM. Most studies did not consider Facial Emotion Recognition (FER) as a part of the ToM construct [references 15, 77-79], so that was not the scope of this review. This information has been included as a potential limitation of the study (p.11, lines 385-388):

‘On the other hand, we did not include the study of Facial Emotion Recognition (FER, previously included in few ToM studies), but this is not considered a part of the ToM construct as reported previously in the scientific literature [15, 77-79].

  1. Henry, J. D.; von Hippel, W.; Molenberghs, P.; Lee, T.; Sachdev, P. S. Clinical assessment of social cognitive function in neurological disorders. Nature Reviews Neurology. 2016, 12, 28-39. https://doi.org/10.1038/nrneurol.2015.229

  1. Yang, D. Y.-J.; Rosenblau, G.; Keifer, C.; Pelphrey, K. A. An integrative neural model of social perception, action observation, and theory of mind. Neuroscience & Biobehavioral Reviews, 2015, 51, 263-275. https://doi.org/10.1016/j.neubiorev.2015.01.020

  1. Péron, J.; Dondaine, T.; Le Jeune, F.; Grandjean, D.;Vérin, M. Emotional processing in Parkinson’s disease: A systematic review: Emotion and PD. Movement Disorders, 2012, 27, 186-199. https://doi.org/10.1002/mds.24025

  1. Argaud, S.; Vérin, M.; Sauleau, P.; Grandjean, D. Facial emotion recognition in Parkinson’s disease: A review and new hypotheses: Facial Emotions and PD: A Review. Movement Disorders, 2018, 33, 554-567. https://doi.org/10.1002/mds.27305

Regarding empathy, affective empathy has been studied in PD, but it is not considered part of ToM [12]. On the other hand, cognitive empathy is like affective ToM, as described in p. 2 lines 83-85 [12,15]:

‘Some authors considered that affective ToM is akin to cognitive empathy, while affective empathy refers to the emotional response to the perceived situation of another [12, 15, 32].

  1. Coundouris, S. P.; Adams, A. G.; Henry, J. D. Empathy and theory of mind in Parkinson’s disease: A meta-analysis. Neuroscience & Biobehavioral Reviews. 2020, 109, 92-102. https://doi.org/10.1016/j.neubiorev.2019.12.030

  1. Henry, J. D.; von Hippel, W.; Molenberghs, P.; Lee, T.; Sachdev, P. S. Clinical assessment of social cognitive function in neurological disorders. Nature Reviews Neurology. 2016, 12, 28-39. https://doi.org/10.1038/nrneurol.2015.229

  1. Bartochowski, Z.; Gatla, S.; Khoury, R.; Al-Dahhak, R.; Grossberg, G.T. Empathy changes in neurocognitive disorders: a review. Ann. Clin. Psychiatry. 2018, 30, 220–232

  1. Also, it would be meaningful to compare ToM deficits noted in PD to other diseases like those in ALS, or MS, as the underlying circuitry affected is different in all these conditions.

We thank the reviewer for this suggestion. We have added to the introduction the following information regarding ToM deficits in ALS and MS [references 22-27(p. 2 lines 60-75].

Later, ToM was investigated in healthy population, schizophrenia [19,20] and in neurodegenerative disorders that share the label of ‘frontotemporal dysfunction syndromes’ such as frontotemporal dementia (FTD) [21] and more recently in amyotrophic lateral sclerosis (ALS) [22]. In this sense, ALS patients, show problems in both affective and cognitive ToM (two parts of ToM explained in more depth below) even in early stages of the disease [23]. Affective ToM deficits in ALS have been related to the disruption of the frontotemporoparietal network [24,25], while cognitive ToM deficits have been linked to a separate network. In this regard, one study found bilateral hypometabolism in the dorsomedial and dorsolateral prefrontal cortex and in the supplementary motor area associated with poor cognitive ToM performance in this disease [26]. Likewise, moderate impairments in overall ToM have also been found in multiple sclerosis (MS) an immune-mediated demyelinating disease of the central nervous system (with no differences between it subtypes). This impairment might be explained by a white matter damage in MS [27]. In general, all these data support the idea that this construct may also be an acquired disorder and is not only developmental.

  1. Benbrika, S.; Desgranges, B.; Eustache, F.; Viader, F. Cognitive, Emotional and Psychological Manifestations in Amyotrophic Lateral Sclerosis at Baseline and Overtime: A Review. Frontiers in Neuroscience, 2019, 13, 951. https://doi.org/10.3389/fnins.2019.00951

  1. Trojsi, F.; Siciliano, M.; Russo, A.; Passaniti, C.; Femiano, C.; Ferrantino, T.; De Liguoro, S.; Lavorgna, L.; Monsurrò, M. R.; Tedeschi, G.; Santangelo, G. Theory of Mind and Its Neuropsychological and Quality of Life Correlates in the Early Stages of Amyotrophic Lateral Sclerosis. Frontiers in Psychology, 2016, 7. https://doi.org/10.3389/fpsyg.2016.01934

  1. Cerami, C.; Dodich, A.; Canessa. N.; Crespi, c.; Iannaccone, S.; Corbo, M.; Lunetta, C.; Consonni, M.; Scola, E.; Falini, A.;  Cappa.; S.F. Emotional empathy in amyotrophic lateral sclerosis: a behavioural and voxel-based mor­phometry study. Amyotroph Lateral Scler Frontotemporal Degener. 2014, 15, 21–29. https://doi.org/10.3109/21678421.2013.785568

  1. Trojsi, F.; Di Nardo, F.; Santangelo, G.; Siciliano, M.; Femiano, C.; Passaniti, C.; Caiazzo, G.; Fratello, M.; Cirillo, M.; Monsurrò, M. R.; Esposito, F.; Tedeschi, G. Resting state fMRI correlates of Theory of Mind impairment in amyotrophic lateral sclerosis. Cortex, 2017, 97, 1-16. https://doi.org/10.1016/j.cortex.2017.09.016

  1. Carluer, L.; Mondou, A.; Buhour, M.-S.; Laisney, M.; Pélerin, A.; Eustache, F.; Viader, F.; Desgranges, B. Neural substrate of cognitive theory of mind impairment in amyotrophic lateral sclerosis. Cortex, 2015, 65, 19-30. https://doi.org/10.1016/j.cortex.2014.12.010

  1. Lin, X.; Zhang, X.; Liu, Q.; Zhao, P.; Zhong, J.; Pan, P.; Wang, G.; Yi, Z. Empathy and Theory of Mind in Multiple Sclerosis: A Meta-Analysis. Frontiers in Psychiatry. 2021, 12, 628110. https://doi.org/10.3389/fpsyt.2021.628110

Reviewer 2 Report

In introduction I suggest a direct description of the cognitive aspects of parkinson's disease. (https://www.nature.com/articles/nrneurol.2017.27) perhaps referring to this recent manuscript instead of [4].

Despite being a narrative review of the literature to provide clarity to the reader I suggest merging the first 3 paragraphs into an introductory section to ensure a rational background and goal.

Figure 1 Add anatomical references of the telencephalon and basal ganglia. The figure is too bare.

234. The term "Correlation" in a review may be inappropriate. I would suggest merging paragraphs 6 and 7 and shortening them.  I wish there was (being a Review) also an evaluation of the paper quantity/quality, the time frame and how the authors approached the problem regarding the MP.

Finally, I suggest a substantial addition of manuscript limitations.

Although it is well written, I believe it is necessary to remove some statements to make the manuscript smoother and more accessible to the reader.

Author Response

Reviewer 2. 

1. In introduction I suggest a direct description of the cognitive aspects of parkinson's disease. (https://www.nature.com/articles/nrneurol.2017.27) perhaps referring to this recent manuscript instead of [4].

We thank the reviewer for this suggestion. We expanded the description of the cognitive aspects of Parkinson´s disease [references 4-8] and referred to the suggested reference that could be more up-to-date [reference 9, pp.1 and 2, introduction part]):

Non-motor symptoms are common in Parkinson`s disease (PD) [1]. In this regard, cognitive impairment and dementia are frequent entities, especially in patients with long disease duration [2,3]. A wide variety of cognitive deficits have been reported in PD, even early in the disease and with different development over time. Classically, the cognitive deficit profile in PD has been linked to a pattern with more executive deficits caused by dopaminergic fronto-striatal network dysfunction in early stages [4]. Nevertheless, patients exhibiting posterior cortical deficits show a higher risk of conversion to dementia [5]. There is also a syndrome representing a stage between normal aging and dementia in PD without loss of functionality (mild cognitive impairment in PD- MCI-PD) [5,6,7], and this is considered a risk factor for dementia [8]. When PD patients develop dementia, they show a greater degree and diversity of cognitive deficits (affecting aspects such as memory, visuospatial function, attention and executive function) severe enough to affect the daily life routine [4]. PD patients can also suffer from subjective cognitive decline (SCD)- a stage that represents an increased risk for cognitive decline in some studies- where cognitive deficits can be noticed by the patient, family or clinician, but might be too slight to be noticed in a formal cognitive evaluation [9].

  1. Goldman, J. G.; Williams-Gray, C.; Barker, R. A.; Duda, J. E.; Galvin, J. E. The spectrum of cognitive impairment in Lewy body diseases: Cognitive Impairment in Lewy Body Diseases. Mov Disord. 2014, 29, 608-621. https://doi.org/10.1002/mds.25866

  1. Goldman, J. G.; Holden, S. K.; Litvan, I.; McKeith, I.; Stebbins, G. T.; Taylor, J.-P. Evolution of diagnostic criteria and assessments for Parkinson’s disease mild cognitive impairment: Evolution Criteria Assessments PD-MCI. Mov Disord. 2018, 33, 503-510. https://doi.org/10.1002/mds.27323

  1. Emre, M.; Aarsland, D.; Brown, R.; Burn, D. J.; Duyckaerts, C.; Mizuno, Y.; Broe, G. A.; Cummings, J.; Dickson, D. W.; Gauthier, S.; Goldman, J.; Goetz, C.; Korczyn, A.; Lees, A.; Levy, R.; Litvan, I.; McKeith, I.; Olanow, W.; Poewe, W.; … Dubois, B. Clinical diagnostic criteria for dementia associated with Parkinson’s disease. Mov Disord. 2007, 22, 1689-1707. https://doi.org/10.1002/mds.21507

  1. Gasca-Salas, C.; Estanga, A.; Clavero, P.; Aguilar-Palacio, I.; González-Redondo, R.; Obeso, J. A.; Rodríguez-Oroz, M. C. Longitudinal Assessment of the Pattern of Cognitive Decline in Non-Demented Patients with Advanced Parkinson’s Disease. Journal of Parkinson’s Disease, 2014, 4, 677-686. https://doi.org/10.3233/JPD-140398

  1. Hoogland, J.; Boel, J. A.; de Bie, R. M. A.; Geskus, R. B.; Schmand, B. A.; Dalrymple-Alford, J. C.; Marras, C.; Adler, C. H.; Goldman, J. G.; Tröster, A. I.; Burn, D. J.; Litvan, I.; Geurtsen, G. J.; on behalf of the MDS Study Group “Validation of Mild Cognitive Impairment in Parkinson Disease”. Mild cognitive impairment as a risk factor for Parkinson’s disease dementia: MCI as a Risk Factor For PDD. Mov Disord. 2017, 32, 1056-1065. https://doi.org/10.1002/mds.27002

  1. Aarsland, D.; Batzu, L.; Halliday, G. M.; Geurtsen, G. J.; Ballard, C.; Ray Chaudhuri, K.; Weintraub, D. Parkinson disease-associated cognitive impairment. Nature Reviews Disease Primers. 2021, 7, 47. https://doi.org/10.1038/s41572-021-00280-3
  2. Despite being a narrative review of the literature to provide clarity to the reader I suggest merging the first 3 paragraphs into an introductory section to ensure a rational background and goal.

We have taken into account the reviewer’s suggestion and merged the introduction as a single paragraph.

  1. Figure 1 Add anatomical references of the telencephalon and basal ganglia. The figure is too bare.

Thank you for this suggestion. We have added to the figure the requested information. 

  1. The term "Correlation" in a review may be inappropriate. I would suggest merging paragraphs 6 and 7 and shortening them.  I wish there was (being a Review) also an evaluation of the paper quantity/quality, the time frame and how the authors approached the problem regarding the MP.

We have merged paragraphs 6 and 7 in 6.1 and 6.2, in order to facilitate the lecture, as follows (pp. 7, 8 and 9):

6.1. Main findings and potential confounders

6.2 Relationship with other cognitive functions and general cognitive function

Furthermore, we have changed the term ‘correlation’ to others more suitable (association and relations), in paragraph 6.2. (from line 268 to 303).

As reported in the limitations (p 11, lines 381-385), this is not a systematic review, therefore, we did not include an evaluation of the paper quantity/quality and the time frame.

On the other hand, we are not aware about what the reviewer means by MP. We are sorry about this inconvenience. We will be happy to address this issue when this is clarified. 

  1. Finally, I suggest a substantial addition of manuscript limitations.

We have added some limitations to the manuscript in lines 381–388 as follow:

There are some limitations in our review. Although we added the most relevant publications of ToM in PD, this is not a systematic review and this prevents us from performing an analysis of paper quantity/quality. Nonetheless, we aimed to report a narrative review including other aspects such as the comprehensive background to the ToM concept, suggested circuitries affected in PD or test descriptions. On the other hand, we did not include the study of Facial Emotion Recognition (FER, previously included in few ToM studies), but this is not considered a part of the ToM construct as reported previously in the scientific literature [15, 77-79].

  1. Henry, J. D.; von Hippel, W.; Molenberghs, P.; Lee, T.; Sachdev, P. S. Clinical assessment of social cognitive function in neurological disorders. Nature Reviews Neurology. 2016, 12, 28-39. https://doi.org/10.1038/nrneurol.2015.229

  1. Yang, D. Y.-J.; Rosenblau, G.; Keifer, C.; Pelphrey, K. A. An integrative neural model of social perception, action observation, and theory of mind. Neuroscience & Biobehavioral Reviews, 2015, 51, 263-275. https://doi.org/10.1016/j.neubiorev.2015.01.020

  1. Péron, J.; Dondaine, T.; Le Jeune, F.; Grandjean, D.;Vérin, M. Emotional processing in Parkinson’s disease: A systematic review: Emotion and PD. Movement Disorders, 2012, 27, 186-199. https://doi.org/10.1002/mds.24025

  1. Argaud, S.; Vérin, M.; Sauleau, P.; Grandjean, D. Facial emotion recognition in Parkinson’s disease: A review and new hypotheses: Facial Emotions and PD: A Review. Movement Disorders, 2018, 33, 554-567. https://doi.org/10.1002/mds.27305
  2. Although it is well written, I believe it is necessary to remove some statements to make the manuscript smoother and more accessible to the reader.

We thank the reviewer for these comments. We modified the manuscript according to the reviewer’s suggestions. If the reviewer still considers any statement needs to be reviewed we will be happy to modify it.

Reviewer 3 Report

Your manuscript is a very well written and comprehensive scholarly review of the current state of research related to the theory of mind of patients with Parkinson's disease. The topic is clearly important from multiple aspects, including patient care and general neurobiological understanding of cognitive systems.

The underlying hypothesis is clearly stated and well conveyed, including an important neuroanatomical circuit figure to help understand the different processes related to the basal ganglia circuitry. Consider providing additional citations for the reader generally interested in such circuitry, perhaps such as: https://europepmc.org/article/NBK/nbk537141 

Your description of neuroimaging studies and their related test performance outcome measures is good. Consider whether you can include a concise description of the ToM-related outcome measures within Table 1, as this information is not readily presented. However, your description of the clinical tests related to ToM are very helpful. 

While potentially outside of your intended scope, consider discussing the role of norepinephrine loss and locus ceruleus degeneration in PD, as well as the connections of the dopaminergic fibers to the cortex and hippocampus from the substantia nigra. 

Overall, your conclusion is well supported and the potential impact of your study on future research is clear.

Author Response

Reviewer 3. 

1.Your manuscript is a very well written and comprehensive scholarly review of the current state of research related to the theory of mind of patients with Parkinson's disease. The topic is clearly important from multiple aspects, including patient care and general neurobiological understanding of cognitive systems.

We thank the reviewer for this comment.

  1. The underlying hypothesis is clearly stated and well conveyed, including an important neuroanatomical circuit figure to help understand the different processes related to the basal ganglia circuitry. Consider providing additional citations for the reader generally interested in such circuitry, perhaps such as: https://europepmc.org/article/NBK/nbk537141

We thank the reviewer for the suggestion. We have added that reference [44] in p.3 line 124 as follows:

‘The dopamine depletion characterizing PD impacts the basal ganglia networks involving not only the motor circuit [44] but also the associative and the limbic one [45] (see Figure 1). 

  1. Young C, B.; Reddy, V.; Sonne J. Neuroanatomy, Basal Ganglia. StatPearls Publishing. [Updated 2021 Jul 31]. 2021 Jan-Available from: https://www.ncbi.nlm.nih.gov/books/NBK537141/
  2. Your description of neuroimaging studies and their related test performance outcome measures is good. Consider whether you can include a concise description of the ToM-related outcome measures within Table 1, as this information is not readily presented. However, your description of the clinical tests related to ToM are very helpful. 

We have included the descriptions of the ToM tests results to table 1. Furthermore, we have added the two scores of the RMET test in p. 7 lines 202-205 as follows:

Two scores can be obtained: the emotion score (the total of items well identified by the patient) and the gender score (by attributing correctly the gender of the photography actor). A percentage of the correct responses can be later calculated in both cases.

  1. While potentially outside of your intended scope, consider discussing the role of norepinephrine loss and locus coeruleus degeneration in PD, as well as the connections of the dopaminergic fibers to the cortex and hippocampus from the substantia nigra. 

We appreciated these suggestions.

Even though, the mesolimbic dopaminergic pathway (where the hippocampus is involved) can be impaired in PD patients [45] and the hippocampus plays a role in social behavior (in general population studies) (see Rubin et al. Frontiers in Human Neuroscience 2014), there are no previous PD studies linking this structure with ToM. As per cortical areas, the limbic and associative basal ganglia circuits that may underlie affective and cognitive ToM deficits respectively include several frontal and cingulate areas (fig. 1, ref. 45 and 43)

  1. Abu-Akel, A. The neurochemical hypothesis of ‘theory of mind’. Medical Hypotheses, 2003, 60, 382-386. https://doi.org/10.1016/S0306-9877(02)00406-1
  2. Obeso, J. A.; Rodríguez-Oroz, M. C.; Benitez-Temino, B.; Blesa, F. J.; Guridi, J.; Marin, C.; Rodriguez, M. Functional organization of the basal ganglia: Therapeutic implications for Parkinson’s disease: Therapeutic Implications for Parkinson’s Disease. Mov Disord, 2008, 23, S548-S559. https://doi.org/10.1002/mds.22062

We have added information regarding norepinephrine (or noradrenaline) loss and locus coeruleus degeneration in PD as follows (p 10, lines 375-380):

Finally, there is a potential relationship between noradrenaline and ToM in the general population [75] and this neurotransmitter is reduced in PD [9]. Noradrenaline relates to cognitive dysfunction (especially with executive dysfunction and attention) [9] as well as depression and apathy in PD [76]. However, no studies have shown a link between norepinephrine loss and locus coeruleus degeneration with ToM deficits in this neurodegenerative disease.

  1. Aarsland, D.; Batzu, L.; Halliday, G. M.; Geurtsen, G. J.; Ballard, C.; Ray Chaudhuri, K.; Weintraub, D. Parkinson disease-associated cognitive impairment. Nature Reviews Disease Primers. 2021, 7, 47. https://doi.org/10.1038/s41572-021-00280-3

  1. Corbetta, M.; Patel, G.; Shulman, G. L. The Reorienting System of the Human Brain: From Environment to Theory of Mind. Neuron, 2008, 58, 306-324. https://doi.org/10.1016/j.neuron.2008.04.017

  1. Remy, P.; Doder, M.; Lees, A.; Turjanski, N.; Brooks, D. Depression in Parkinson’s disease: Loss of dopamine and noradrenaline innervation in the limbic system. Brain, 2005, 128, 1314-1322. https://doi.org/10.1093/brain/awh445
  2. Overall, your conclusion is well supported and the potential impact of your study on future research is clear.

We thank again the reviewer for the kind comment.

Round 2

Reviewer 2 Report

I must congratulate that the manuscript seems enriched. Thank you for having better structured the introduction.

66. This insightful paragraph could go further in addition to the ALS and the MS with PLS mimicking parkinson disease. I provide you a bibliographic reference for completeness https://doi.org/10.3233/nre-201527

Author Response

Reviewer comment:

  1. This insightful paragraph could go further in addition to the ALS and the MS with PLS mimicking parkinson disease. I provide you a bibliographic reference for completeness https://doi.org/10.3233/nre-201527

We thank the reviewer for the positive comments and for this suggestion. However, we consider this case controversial as the diagnosis of PD did not follow the clinical criteria. PD diagnosis was performed according to the interview. However, the classical and also the more recent proposed diagnostic criteria for PD includes the presence of bradykinesia (+ rigidity or rest tremor) as mandatory clinical finding for the diagnosis of PD (Hughes et al. 1992, doi: 10.1136/jnnp.55.3.18; Postuma et al. 2015, 10.1002/mds.26424). In this case report, there is no mention about a clinical exam suggesting PD. In addition, the normal DaTSCAN did not support this diagnosis.

It is likely that the impairment of handwriting and fine motor skills of hands in 2015 (and similarly difficulty in climbing stairs and feeling of weakness in legs in 2016)  would be related to weakness and spasticity. Therefore, we cannot rule out that the patient had any upper motor dysfunction (including signs such as spasticity, weakness and/or pathological hyperreflexia) leading to PLS diagnosis (Turner et al. 2020, doi:10.1136/jnnp-2019-322541). Therefore, we do not consider this case illustrative.  Also, we cannot find any information about theory of mind performance in this patient.